# Cross-Cultural Adaptation and Validation of the Pregnancy Mobility Index for the Italian Population: A Cross-Sectional Study

**DOI:** 10.3390/healthcare10101971

**Published:** 2022-10-08

**Authors:** Andrea Manzotti, Sonia Zanini, Sofia Colaceci, Niccolò Giovannini, Agnese Antonioli, Alice Ziglioli, Francesco Frontani, Giovanni Galeoto

**Affiliations:** 1Division of Neonatology, “V. Buzzi” Children’s Hospital, ASST-FBF-Sacco, 20154 Milan, Italy; 2RAISE Lab, Clinical-Based Human Research Department, Foundation COME Collaboration, 65121 Pescara, Italy; 3Research Department, SOMA, Istituto Osteopatia Milano, 20126 Milan, Italy; 4Departmental Faculty of Medicine and Surgery, Saint Camillus International University of Rome and Medical Sciences (UniCamillus), 00131 Rome, Italy; 5Department of Obstetrics and Gynecology, Fondazione IRCCS Ca’ Granda Ospedale Maggiore Policlinico, 20122 Milan, Italy; 6Department of Human Neurosciences, Sapienza University of Rome, Viale delle Università 30, 00185 Rome, Italy; 7IRCSS Neuromed, Via Atinense 18, 86077 Pozzilli, Italy

**Keywords:** pregnancy, validation, questionnaire, mobility, Italy

## Abstract

Introduction: Pregnancy is a specific condition that modifies the mobility of women. In this population, it seems important to use specific tools to properly assess them. The Pregnancy Mobility Index (PMI) was created in 2006 with the aim of assessing mobility in pregnant women. The goal of this study was to translate, adapt, and evaluate the statistical properties of the questionnaire in the Italian pregnant population. Methods: The PMI underwent translation and transcultural adaptation. Reliability and concurrent validity, compared to the Oswestry Disability Index (ODI), was investigated on a sample of pregnant women. An ANOVA was performed to detect differences in the PMI score considering the Body Mass Index (BMI) and age of the sample. Results: The PMI was forward translated, back translated, and transculturally adapted. A consensus meeting accepted the final version of the questionnaire. The PMI was given to 93 pregnant women. PMI showed excellent reliability for every item and the total score (Cronbach’s alpha of 0.945). Concurrent validity compared with ODI items 2–9 was strong considering the total score, with r = 0.726, but moderate comparing the first item of the ODI and the total score of the PMI, r = 470, and considering the total score of both questionnaires (r = 0.683). The ANOVA showed statistical difference in pregnant women with lower BMI for every subscale and total score of PMI (*p* = 0.009) and for outdoor mobility considering age (*p* = 0.019). Conclusions: The PMI seems to be a valid and reliable tool to assess mobility in the pregnant population. Pregnant women with a lower BMI showed a greater mobility score in the PMI. In turn, younger pregnant women presented a lower mobility score compared to older pregnant women.

## 1. Introduction

Pregnancy puts great demands on the body of a woman, posing a psychic, somatic, and often also social burden [1].In fact, the physical changes during pregnancy could affect the daily activities of pregnant women, reducing their quality of life. Limitations due to physical changes and the fear of managing labour are shown to be the riskiest areas that could affect pregnant women [1]. Moreover, employed pregnant women seem to feel pregnancy as interfering with their work and household tasks due to a reduction in mobility [2]. It is well known that pregnancy modifies the musculoskeletal system in different ways. The weight gain changes the joints’ loading pattern and exaggerates lumbar lordosis; in turn, the hormones contribute to increased ligamentous laxity, particularly in the pelvis [3].These mechanisms result in increased pressure on the lumbosacral spine and sacroiliac joint, causing sacroiliac dysfunction [4].These modifications could lead to an increase of suffering from low back pain (LBP) and pelvic girdle pain (PGP), as stated by Kanakaris et al. and Liddle and Pennick [5,6]. PGP leads to difficulties in everyday life activities, increased stress levels, lower engagement in social life, decreased efficiency at school or at work, and lowered sexual life satisfaction [7,8]. When examining patients with musculoskeletal pain in clinical practice, there is a focus on identifying the underlying mechanisms of the pain (pathophysiology) and the functional consequences of having pain in order to be able to provide appropriate care [9]. Moreover, it is well recognized that activity is fundamental during pregnancy because less active or unactive pregnant women could be affected by different disorders, such as gestational diabetes, low back pain, sleep disorders, and hypertensive disorders [10,11,12,13]. To assess the pain and mobility during pregnancy, there is a specific condition questionnaire, the Pelvic Girdle Questionnaire (PGQ), that has been translated and adapted in many languages but not in Italian. However, this questionnaire does not specifically address mobility during pregnancy nor considers the potential level of disability; consequently, PGP also seems fundamental to assess the mobility of the patients, to clearly define their health status during the assessment. Considering these factors, it seems fundamental to have a specific tool to properly assess the different demands of the pregnant population, and the Pregnancy Mobility Index (PMI) seems to accurately evaluate this particular population [14]. PMI was specifically created to assess the mobility of pregnant woman. PMI is a specific condition tool fundamental for clinicians who would like to investigate the modification related to an intervention to increase mobility or to analyze and identify categories of patients that could be at risk of a low level of mobility during pregnancy. Moreover, the PMI is an inexpensive tool for the clinician to use to investigate the mobility of pregnant women. Nowadays, the Oswestry Disability Index (ODI) and Numeric Rating Scale (NRS) are the most common disability tools adopted to evaluate PGP [15,16,17]. However, these tools are not specific for pregnant women and, moreover, not validated for this population. The goal of this study is to translate, cross-culturally adapt, and evaluate the psychometric properties of the PMI in a pregnant population; in this way, it will be possible to adopt this instrument to evaluate pregnant women in Italy.

## 2. Methods

The first step was to receive the consent from the developers of the original instrument, and then the PMI was translated from English to Italian using the “Translation and Cultural Adaptation of Patient Reported Outcomes Measures—Principles of Good Practice” guidelines [18]. The reliability and validity of the culturally adapted scale were assessed following the “Consensus Based Standards for the Selection of Health Status Measurement Instruments” (COSMIN) checklist [19]. As suggested by Beaton, it is fundamental to translate, adapt, and evaluate the statistical properties of this questionnaire in the Italian population in order to use it properly in a clinical setting [20]. All procedures followed were in accordance with the ethical standards of the responsible committee on human experimentation (institutional and national) and with the Helsinki Declaration of 1975, as revised in 2008. Ethics committee approval was not required for this study; this research involved secondary use of clinical data, which are provided without any identifier or group of identifiers, which would allow attribution of private information to an individual. Informed consent was obtained from all participants for being included in the study.

### 2.1. Translation and Cultural Adaptation

During the first stage (translation process), two researchers proceeded with a forward translation. The translators produced independent versions of the questionnaire, which were compared, and the results were synthesized by AM. We contacted a professional translator, but this was not helpful because both versions retrieved during the first translation process were similar. Without having seen the original version, two other researchers back translated into the original language the final version of the forward translation. The back-translated version of the instrument was compared with the original. After this, A.M., who were familiar with both English and Italian, reviewed the first translated version, and then reworded and reformulated some items to minimize any differences from the original version, and to properly adapt the translated version into the Italian culture. Last part of the process was a consensus meeting between the authors to draft the final version of the questionnaire.

#### 2.1.1. Participants

A sample size of pregnant women was recorded from gynaecologists during the first obstetric visit in the prenatal outpatient department in Clinica Mangiagalli in Milan. Two investigators were responsible for data collection under the supervision of the principal investigator. During the study, a researcher experienced in data quality performed a periodic check of the collected data to verify its consistency. The Mangiagalli Clinic is a public university maternity hospital located in Milan, which is in second place in Italy by volume of activity. Women in both the groups provided written consent for participation in the study. They had to fill out the questionnaires at baseline, during the first obstetrician visit at the end of the first trimester, and at follow-up; so, in the last assessment before the delivery between weeks 38 and 40 of pregnancy. The allocations were archived in a document protected by password on the group’s research computer. Moreover, sample characteristics, such as smoking, BMI, age, and type of pregnancy, were recorded. These characteristics were helpful to retrieve statistical analysis and to identify the subgroup of population in which the scale could be administered in different way or considering differently the results. Two investigators were responsible for data collection under the supervision of the principal investigator. During the study, a researcher experienced in data quality performed a periodic check of the collected data to verify its consistency. Inclusion criteria for participation in the study were nulliparous women, singleton pregnancies, spontaneous pregnancies, gestational age between the 12th and 14th week, no maternal medical conditions, maternal age between 18 and 48 years, and no language barrier. Exclusion criteria were multiple pregnancies, in vitro fertilization pregnancies, maternal or foetal medical conditions, age < 18, and language barrier. Paper data were archived in a protected area at Clinical Mangiagalli. The recruitment for this study lasted from February 2019 to December 2021. 

#### 2.1.2. Questionnaire

The Pregnancy Mobility Index (PMI) is a questionnaire composed of 24 items. The questionnaire is composed of three sub scales: the first one considers the daily mobility in the house, and it is composed of 7 items; the second considers normal household activity and is composed of 9 items; and the third considers the mobility outdoors and is composed of 8 items. The PMI is a specific condition self-assessed questionnaire and for every item women were asked if they suffered from PGP while performing the activities by choosing one of four possible answer: ‘no problems performing this task’; ‘some effort performing this task’; ‘much effort performing this task’; and ‘performing this task is impossible or only possible with the aid of others’. Every answer corresponds to a score from 0 to 3. A higher score indicates more severe disability. The score is made by calculating the mean value of the items of that scale. The PMI shows an excellent internal consistency (Cronbach’s alpha = 0.8 to 0.9) and is adequate in detecting change in mobility through pregnancy (van de Pol et al., 2006).

The Oswestry Dysability Index (ODI) consists of 10 items with scores from 0 to 5, where a higher score represents a greater extent of disability. The first section rates the intensity of pain, and the remaining ones cover the disabling effect of pain on typical daily activities: personal care, lifting, walking, sitting, standing, sleeping, sex life, social life, and traveling. The questionnaire is typically completed in about 5 min and scored in less than 1 min [21]. It was validated in the Italian language from Monticoni et al. 2009 [22]. All the instruments were administered at baseline, during the first obstetrician visit at the end of the first trimester (12–14 week).

### 2.2. Statistical Analysis

The IBM^®^ SPSS^®^ version 27 tool (Chicago, IL, USA) was used to carry out the statistical analysis. 

#### 2.2.1. Reliability

The internal consistency was defined as the extent to which items in a subscale are homogeneous, thus measuring the same concept [23]. Internal consistency of the PMI was examined by Cronbach’s alpha (α), to assess homogeneity of the scale by measuring the interrelatedness of the items. Two subscale scores were calculated separately at baseline assessment. A Cronbach’s α between 0.70 and 0.95 was considered satisfactory [23]. 

#### 2.2.2. Construct Validity

Construct validity was assessed using the Pearson’s correlation analysis to determine the association between the Italian version of the ODI and the three subscales of the PMI. We hypothesized that a high degree of confidence between these two questionnaires were present. Coefficients < 0.30, 0.30 to 0.60, and >0.60 were considered to indicate low, moderate, and high correlations, respectively [24]. 

#### 2.2.3. Cross-Cultural Analysis

To detect if there is a demographic characteristic of the sample that could modify the score of the questionnaire, we performed an ANOVA test for more samples. The *p*-value was set at 0.05. We included in the analysis the three subscales of the PMI and the total score, matched with the age and the Body Mass Index (BMI) of the pregnant women. In this way we would like to identify if there is a characteristic of the pregnant women that could modify the score of the questionnaire.

## 3. Result

### 3.1. Translation and Cultural Adaptation

The final version of the PMI was drafted after a consensus meeting between the authors. The consensus meeting was made after the forward translation, back translation, and cultural adaptation process. The final version of the PMI was carried out and all the items were similar to the original version. So, we administered this version of the questionnaire to the sampled population included in the paper. The only modification item that was debated was item 8, “vacuum cleaning”. This item in Italian was translated as “cleaning on the floor with a vacuum cleaner”.

#### Participants

The recruitment for this study lasted from February 2019 to December 2021. The participants were 93 pregnant women with an age range between 20 and 48 (mean age 34.55 SD ± 4.75). The demographic characteristics of the sample are summarized in Table 1.

### 3.2. Reliability

The PMI was found to have an excellent item’s Cronbach alpha, which ranges from 0.943 to 0.949, and a Cronbach score with alpha deleted of 0.816 to 0.903, considering all the items. The three subscale scores showed an excellent reliability in every subscale: daily mobility in the house, alpha = 0.946; household activities, alpha = 0.947; mobility outdoors, alpha = 0.946; and total score, 0.945. The alphas for every item are shown in Table 2.

#### 3.2.1. Construct Validity

The Pearson’s correlation was high considering the total score and the subscales of the PMI, compared with items 2–9 of the ODI, which are part of the function subscale. The score ranges from r = 648 to r = 726.The concurrent validity was found moderate considering ODI item 1, the one which considered the pain of pregnant women, with a score that ranges from r = 0.401 to r = 0.484. The comparison between the total score of the ODI and PMI showed a high correlation, with a score that ranges from r = 0.32 to r = 0.683 (Andresen, 2000). In Table 3, the r for every subscale and total score are shown.

#### 3.2.2. Cross-Cultural Analysis

A cross-cultural analysis was performed to detect if the score of the PMI could differ in a subgroup of the population. To analyse the cross-cultural differences, an ANOVA was performed, and a statistical difference was found for BMI in every subscale and the total score (*p* = 0.009), and for the mobility outdoors considering the age (*p* = 0.019). In Table 4, the ANOVA analysis is shown. 

## 4. Discussion

The goal of this study was to translate, adapt, and evaluate the psychometric characteristics of the PMI in the Italian language. The Cronbach’s alpha was found to be high considering all the items (alpha deleted = 0.816 to 0.903), the three different subscales of the PMI (mobility outdoor, alpha = 0.946; household activities, alpha = 0.947; and daily activity in the house, alpha = 0.946), and also the total score of the questionnaire, showing a high Cronbach’s alpha (0.945). These results are consistent with the paper from van de pol et al., who created and evaluated the statistical properties of this specific tool (van de Pol et al., 2006). The PMI considers mobility during pregnancy, so we decided to evaluate the concurrent validity using the ODI scale. Although the ODI is not validated for pregnant populations and some items of this questionnaire does not fit with a specific population, such as pregnant women (items 7-8-10), the ODI has been used by other authors to investigate the concurrent validity for other specific condition questionnaires in pregnant populations, such as the PGQ [25,26,27,28,29].Considering the items of the PMI, we investigated the correlation with the ODI items regarding function (items 2 to 9). The correlation between the two scales was found strong considering the total score (r = 0.726). Comparing the PMI subscales, the correlation was found moderate, with an r = 0.648 considering the daily mobility in the house; r = 0.687 considering the household activities; and r = 0.669 considering the mobility outdoors. So, the PMI could be considered as a specific tool to assess mobility in pregnant women. Considering the different needs among the pregnant population seems fundamental to have specific, validated tools to assess this peculiar population. Interesting were the result of the ANOVA. In fact, in our sample, at baseline between the first and second trimester of gestation (between 12 and 14 weeks of gestational age), the BMI seems to modify in a significant way the score of the PMI. Pregnant women with a lower BMI showed a higher score on the questionnaire, meaning that these pregnant women suffer from a higher reduction in mobility. These results were statistically significant for every subscale of the PMI and for the total score (*p* = 0.009). Usually, a higher BMI is considered a risk factor for PGP and a reduction in mobility during pregnancy, but authors agree that multiple factors should be addressed when considering pain and mobility in pregnancy [30,31,32,33]. So, these results could be interesting and further analysis could be taken into consideration. In fact, Lisonkova et al. demonstrate that a lower BMI and higher BMI level could be equally considered as risk factors for morbidity in pregnancy [34]. In the future, considering an assessment that matches specific tools such as the PMI with other questionnaires that consider the psycho-social aspects of pregnancy could be interesting to understand if during pregnancy between the first and second trimester of gestation (12 to 14 weeks of gestational age) the BMI is an aspects causing a reduction in mobility, and these could be related with beliefs or other psyco-social aspects and not only to the physical changes of the women, as is suggested by Bakker et al. [35]; this analysis, however, is beyond the scope of this paper, so we are not going to discuss it further. The ANOVA analysis also showed a difference considering the mobility outdoors subscales. In this section, the younger women, 25–29 years old, obtained a better score than older pregnant women. This analysis is in contrast with the literature, which suggest that a younger age could be a risk factor for developing PGP [31,32].However, the small sample—just 4 participants in this group—does not allow to generalize these findings. Unfortunately, it was not possible to compare our results with other studies because we did not find other studies that assessed the psychometric properties of the PMI. Further analysis could be necessary to confirm the interesting result from the ANOVA analysis, which showed a greater disability of pregnant women with a lower BMI level, and to investigate the reason behind this difference compared to other BMI levels. To the best of our knowledge, this is the first study to adapt and translate the PMI, which is a questionnaire often used in clinical practice, which could be helpful for clinicians and researchers that would like to investigate the efficacy of a treatment in this specific population. 

## 5. Limitation

Some limitations of the study should be underlined. Data collection was performed in only one centre and the sample is smaller than predicted due to the COVID-19 pandemic, which did not allow us to collect data as wished. Moreover, due to the pandemic, we could not perform the two parts of the validation process, such the pre-test of the final version of the questionnaire and the reliability of the questionnaire. In fact, we expected to collect more data, but considering the anxiety of pregnant women during these times, and the strict rules prescribed to limit COVID-19 spreading, we were not able to perform these two analyses. For the same reasons, some groups of the cross-cultural analysis were too small (BMI > 30, age between 45 and 49). So, the statistical analysis is unpowered. These categories should be analysed in future with a bigger sample.

## 6. Conclusions

PMI seems to be a reliable and valid questionnaire to assess the mobility of pregnant women. The tool showed a good score on reliability, which was found to be excellent, with the total score Cronbach’s alpha = 0.945. The concurrent validity of the PMI with ODI items 2 to 9 achieved a strong correlation score, with an r = 0.726, while the correlation with the first item of the ODI, which considered the pain, was found to be moderate (*p* = 0.470).

## Figures and Tables

**Table 1 healthcare-10-01971-t001:** Demographic characteristics of the 93 participants for validation of the PMI.

	n. = 93
Age Mean ± SD years	34.55 (4.75)
Age, N (%)	
20–24	1 (1.1)
25–29	4 (4.4)
30–34	24 (25.8)
35–39	39 (41.8)
40–44	23 (24.7)
45–49	2 (2.2)
BMI Mean ± SD	23.3 (2.8)
BMI, N (%)	
<18	10 (10.8)
19–24	68 (73.1)
25–29	12 (12.9)
>30	3 (3.2)
Planning pregnancy, N (%)	
Yes	10 (10.7)
No	83 (89.3)
Natural pregnancy, N (%)	
Yes	92 (98.9)
No	1 (1.1)
Smoking, N (%)	
Yes	4 (4.3)
No	79 (95.7)

**Table 2 healthcare-10-01971-t002:** Comparing the total score Cronbach’s alphas of the items and subscales.

Item	Total Score Alpha	Alpha Deleted
@1	0.947	0.863
@2	0.945	0.841
@3	0.948	0.847
@4	0.944	0.824
@5	0.947	0.846
@6	0.949	0.869
@7	0.943	0.833
Subscale 1: Daily mobility in the house	0.946	0.866
@8	0.946	0.901
@9	0.946	0.899
@10	0.946	0.899
@11	0.944	0.887
@12	0.945	0.894
@13	0.946	0.895
@14	0.944	0.880
@15	0.944	0.880
@16	0.944	0.891
Subscale 2: Household activities	0.947	0.903
@17	0.946	0.831
@18	0.946	0.843
@19	0.946	0.858
@20	0.946	0.816
@21	0.947	0.841
@22	0.946	0.828
@23	0.947	0.837
Subscale 3: Mobility outdoors	0.946	0.893
Total score	0.945	

**Table 3 healthcare-10-01971-t003:** Pearson’s correlation scores comparing the PMI with the ODI scale. ** *p* < 0.01.

	Daily Mobility in the House	Household Activities	Mobility Outdoors	Total Score
ODI item 1	0.484 **	0.426 **	0.401 **	0.470 **
ODI item 2–9	0.648 **	0.687 **	0.669 **	0.726 **
ODI total score	0.632 **	0.654 **	0.636 **	0.683 **

**Table 4 healthcare-10-01971-t004:** ANOVA results considering the PMI subscales and total score, and age and BMI of the sample * *p* < 0.05.

Boby Mass Index		n. (%)	Mean ± SD	f	*p* Value
Daily mobility in the house	<18	10 (10.8)	3.70 (4.05)	3.963	0.011 *
	19–24	68 (73.1)	1.9 (2.33)		
	25–29	12 (12.9)	0.93 (1.52)		
	>30	3 (3.2)	5 (1.73)		
Household activities	<18	10 (10.8)	6.10 (5.36)	3.984	0.010 *
	19–24	68 (73.1)	3.31 (4.1)		
	25–29	12 (12.9)	1.17 (1.27)		
	>30	3 (3.2)	7.67 (2.31)		
Mobility outdoor	<18	10 (10.8)	3.7 (4.42)	3.421	0.021 *
	19–24	68 (73.1)	1.4 (2.47)		
	25–29	12 (12.9)	0.25 (0.45)		
	>30	3 (3.2)	1.33 (1.15)		
Total score	<18	10 (10.8)	13.5 (12.9)	4.124	0.009 *
	19–24	68 (73.1)	6.6 (8.22)		
	25–29	12 (12.9)	2.25 (2.7)		
	>30	3 (3.2)	14 (1.73)		
**Age**		**n. (%)**	**Mean ±SD**	**f**	***p* value**
Daily mobility in the house	20–24	1 (1.1)	–	2.081	0.075
	25–29	4 (4.4)	3.75 (4.78)		
	30–34	24 (25.8)	2.08 (2.43)		
	35–39	39 (41.8)	2.36 (2.62)		
	40–44	23 (24.7)	1.17 (1.96)		
	45–49	2 (2.2)	–		
Household activities	20–24	1 (1.1)	–	1.443	0.217
	25–29	4 (4.4)	3.25 (4.3)		
	30–34	24 (25.8)	4.21 (5.2)		
	35–39	39 (41.8)	3.7 (4)		
	40–44	23 (24.7)	2.1 (2.7)		
	45–49	2 (2.2)	2		
Mobility outdoor	20–24	1 (1.1)	–	2.864	0.019 *
	25–29	4 (4.4)	0.25 (0.5)		
	30–34	24 (25.8)	1.96 (3.14)		
	35–39	39 (41.8)	1.5 (2.7)		
	40–44	23 (24.7)	1.04 (1.52)		
	45–49	2 (2.2)	–		
Total score	20–24	1 (1.1)	–	1.967	0.091
	25–29	4 (4.4)	7.25 (9.14)		
	30–34	24 (25.8)	8.25 (10.14)		
	35–39	39 (41.8)	7.5 (8.7)		
	40–44	23 (24.7)	4.3 (5.95)		
	45–49	2 (2.2)	2		

## Data Availability

All the data generated or analyzed during this study are included in this published article.

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
