# Peer review of "Cross-Cultural Adaptation and Validation of the Pregnancy Mobility Index for the Italian Population: A Cross-Sectional Study"

_healthcare, 2022, doi:10.3390/healthcare10101971_

Round 1
Reviewer 1 Report
The manuscript "Cross-Cultural Adaptation and Validation of the Pregnancy Mobility Index for the Italian Population a Cross-Sectional Study" may be an interesting article for Healthcare readers, and the tool can be helpful to other Italian studies. However, some significant concerns should be revised to improve the quality of the manuscript.
Major comments:
a) The PMI is a tool built to assess self-report mobility designed explicitly for a pregnant population. In some parts of the manuscript, the definition of PMI is not it and sometimes is confused with functionality assessment. Please, carefully revise it in the paper.
b) The introduction is unclear; the authors should show a summary of current understanding and background information about the topic, stating the purpose of the work in the form of the research problem. Why is it important to have this tool adapted for Italian pregnant women? The sentence in the introduction about Beaton should be avoided. This information can move to the methodology section.
c) There are some doubts about translation and cultural adaptation in the methodology section. For example, did bilingual professional conduct the first translation from English to Italian? Were the two versions of the translation assets to an expert panel? Was the final version of the questionnaire tested to see if it was understandable by the study population?
d) More detailed information about the sample size calculation and the selection of the participants should be included. In addition, the sentence "A randomization software was used to assign the number 1 to the intervention and 0 to the control group" should be eliminated due to understanding. The authors should include information about the ethical consideration of the study and the number and name of the ethical committee which approved the study.
e) In the description of the questionnaire, it needs to add extra information about the categories of answers to the questions.
f) The information about the statistical analysis to internal consistency, concurrent validity and bivariant analysis should move to statistical analysis. It is needed to add the version of SPSS used to analysis of the data.
g) Why did the authors not test the scale's reliability using test-rests assessment?
h) The author should be included results about the translation and cultural adaptation, for example, indicating the main changes of the questionnaire.
i) Some parts of the text in the first paragraph of the participant's results should move to the study's methodology. Lines 139-143. In table 1, the distribution in categories of age and BMI are not good because some categories have a small number of pregnant women. It can be a problem in the statistical analysis. In addition, the authors should include information about the demographic variables collected in the methodology section. Please define what the meaning of natural pregnancy is.
j) The information in table 2 and the text is very confusing. Please, add Cronbach alfa to the total and the total subscales.
k) In the construct validity, why the PMI total and subscale is not compared to total ODI? Please, add information about this in the methodology section.
l) Table 4 is not clear. The comparison of the PMI and Body mass index in the categorical version is not precise due to in the >30 category; There were three women. This statistical analysis is unpowered. It is like age analysis. Please, modify this analysis for regression models using BMI and age as continuous variables. The figures did not add extra information due to should be eliminated.
m) The discussion should be improved considerably. It needs to add limitations and strengths of the study. In addition, the directions of this error and it can affect the external and internal validity should be included.
n) The conclusion of the study should be improved. The conclusion should be the answer to the study's hypothesis and, in your case, is a repetition of the results. Moreover, the authors did not write association with an ANOVA analysis.
Minor changes
a) The abstract is very confused, and it should be revised. For example, this sentence is quite strong "Pregnant population differs from the general population so analysing the disability in this specific population seems important to properly take care of them". Pregnancy is not a disability or disease. A significant number of women continue with their daily living activities without problems. The discussion should not be added to the abstract.
b) Change the title in the 2.1.3 point to Internal consistency in the methodology section and the 2.2.2 of the results.
c) What do you write in this sentence, "two sub-scales scores were calculated separately at baseline assessment"? The PMI has three subscales.
d) The decimal separator should be changed to "." instead of ",".
e) In this sentence, "The PMI was found to have an excellent item's Cronbach alfa, which ranges from 0.816 to 0.901."
Author Response
We are thankful for your revision, and we tried to improve our work following your indication. An uploaded file was provided with the point-by-point reply.

Reviewer 2 Report
Thank you for your manuscript on the Cross Cultural Adaptation and Validation of the Pregnancy Mobility Index for the Italian Population a Cross Sectional Study. I read the article with great curiosity. I have a few comments:
Line 35 Keywords - I suggest adding the word: Italy.
Line 93: gestational age ≥12 – please add the indication - 12 weeks.
Line 101: The PMI - please provide the full name of the questionnaire.
Line 107: The ODI - please provide the full name of the questionnaire.
Line 100-110 - I suggest that the authors describe the questionnaires a little more precisely.
Line 130 - please describe the Statistical analysis part.
Line 131 – 132 - The IBM® SPSS® tool (Chicago, IL, USA) was used to carry out the statistical analysis. RESULTS - Requires some refinement.
Line 133-137 - This paragraph should not be in the 2.2. Statistical Analysis section.
Line 88 and line 138 - names of subsections are duplicated, which makes them unclear for the reader, they should be refined.
Line 146 - Table 1 - no need to enter height and weight, BMI alone is enough.
If M(SD) stands for the Mean and standard deviation of the total scores, statistically it should be presented as Mean ± SD.
The records of BMI n ° (%) or Planning pregnancy N ° (%) should be standardized (Table 1).
Figure 1-2-3-4-5-6-7-8 - this is a duplication of the results from Table 4.
Discussion - apart from individual items of the literature, there is no citation of the results of other authors. This section is lack of discussions on the main findings reported in Results section.
No consent from the Bioethics Committee to conduct the research - please add.

Author Response
We are thankful for your revision, and we tried to improve our work following your indication. The methods section is the most modified part as you could see from the manuscript. In the following part you will find the point-by-point reply.

Reviewer 3 Report
This is a very interesting study.
The English grammar needs a general revision.
I suggest a few corrections, as follows.
Abstract (example)
The pregnant population differs from the general population, so the analysis of mobility and disability levels in this specific population seems important to properly take care of them. The Pregnancy Mobility Index (PMI) was created in 2006 with the aim of assessing physical function in pregnant women.
BMI and ODI should be explained in advance.
(...) specific tool to assess the physical ability of the pregnant population (...).
(...) greater/lower disability should be replaced by lower/greater scores of mobility (...).
INTRODUCTION
The biomechanical adaptations of gait during pregnancy should also be addressed since they will impact mobility and pain. However, biomechanical analysis systems are expensive and require specific expertise. This is one of the reasons underlying the utility of questionnaires such as this one.
METHODS
What was the gestational age of the participants? gestational age ≥12 was an inclusion criterion, but there might be differences in mobility regarding the 2nd and the 3rd trimester of pregnancy.
What was the intervention? What is the rationale for a control group? Please explain.
Please explain what is and why ODI was used. Construct validity?
(...) between NPRS and the pain sub-scale of the PGQ. (...) should be explained in advance.
(...) we performed an ANOVA test for more sample. (...) Please explain the meaning of this sentence.
(...) matched with the age and the body mass index (BMI) of the pregnant women. (...) What about gestational age?
RESULTS
(...) The final version of the PGQ was drafted after a consensus (...). Please explain in light of the objectives of the study.
(...) During the experimentation (...). Please revise the wording.
Inclusion criteria included maternal age between 18 and 44 years. Table 1 shows 2 participants over 45 years. Were they included or excluded? Please explain.
Was BMI calculated pre-pregnancy? Please explain.
DISCUSSION
"Usually, higher BMI were correlated with more disability on general population (...)". Reference is missing.
"(...) using specific tool such PMI and comparing it to other questionnaire which consider the psyco-social aspects of pregnancy(...)". Reference is missing.
"This could be considered normal on pregnant women. In fact, pregnancy in older women is usually associated with higher level of disability. (...)". Reference is missing.
(...) achieved less disability than (...) should be replaced by "obtained a better score than".
Author Response

(The authors gave the same response as above.)

Round 2
Reviewer 1 Report
Thanks to the authors for answering all my questions satisfactorily. However, there are minor changes that should be revised.
1. The changes conducted in the manuscript should be applied to the abstract.
2. Please change the title "discussion and conclusions" to "conclusions". It is not usual to include discussion in abstracts.
3. In the conclusion of the study in the abstract and in the manuscript the authors said that "lower level of BMI could be associated with higher score on the PMI". However, they did not explore the association with the statistical analysis performed. As indicated in the previous version the ANOVA analysis is not appropriate to explore association. The ANOVA is useful to explore differences in the mean, but no association.
4. The limitations of the study should include previous of the conclusion of the study.
5. The conclusion should be the answer to the study's hypothesis and, in your case, is a repetition of the results. The question of this research would be: is PMI a reliable and valid tool for use in women to assess mobility? Therefore, the conclusion should be the answer to this question.
Author Response
We are thankful for your revision, and we modified the paper as suggested. In the following part you will find the point-to-point answer.
- The changes conducted in the manuscript should be applied to the abstract.
As suggested, we modified the abstract
- Please change the title "discussion and conclusions" to "conclusions". It is not usual to include discussion in abstracts.
As suggested, we removed the terms discussion
- In the conclusion of the study in the abstract and in the manuscript the authors said that "lower level of BMI could be associated with higher score on the PMI". However, they did not explore the association with the statistical analysis performed. As indicated in the previous version the ANOVA analysis is not appropriate to explore association. The ANOVA is useful to explore differences in the mean, but no association.
As suggested, we modified the abstract and conclusion section (line 252).
- The limitations of the study should include previous of the conclusion of the study.
As suggested, we modified the editing of the text
- The conclusion should be the answer to the study's hypothesis and, in your case, is a repetition of the results. The question of this research would be: is PMI a reliable and valid tool for use in women to assess mobility? Therefore, the conclusion should be the answer to this question.
As suggested, we modified the conclusion